# SHAKE-GNN: SCALABLE HIERARCHICAL KIRCHHOFF-FOREST GRAPH NEURAL NETWORK

## ABSTRACT

Graph Neural Networks (GNNs) have achieved remarkable success across a range of learning tasks. However, scaling GNNs to large graphs remains a significant challenge, especially for graph-level tasks. In this work, we introduce SHAKE-GNN, a novel scalable graph-level GNN framework based on a hierarchy of Kirchhoff Forests, a class of random spanning forests used to construct stochastic multi-resolution decompositions of graphs. SHAKE-GNN produces multi-scale representations, enabling flexible trade-offs between efficiency and performance. We introduce an improved, data-driven strategy for selecting the trade-off parameter and analyse the time-complexity of SHAKE-GNN. Experimental results on multiple large-scale graph classification benchmarks demonstrate that SHAKE-GNN achieves competitive performance while offering improved scalability.

## 1 INTRODUCTION

Graph classification is a fundamental task in graph machine learning, in which one aims to determine the class of an entire graph based on its structure and attributes. This task is widely applicable across domains such as molecular property prediction, e.g., identifying active versus inactive compounds (Wu et al., 2021), social network analysis, e.g., bot detection (Zhou et al., 2020), and communication system diagnostics, e.g., faulty topology detection (Zhang et al., 2022). With the rapid advancement of Graph Neural Networks (GNNs), many approaches have achieved impressive results by leveraging local neighbourhood information through message passing (Gilmer et al., 2017). Despite their success, significant challenges remain in terms of scalability and global structure modelling, especially when dealing with large and complex graphs, such as protein graphs and social networks among others.

Early GNNs such as Graph Convolutional Networks (GCNs) (Kipf & Welling, 2017), Graph Attention Networks (GATs) (Veličković et al., 2018), and Graph Isomorphism Networks (GINs) (Xu et al., 2019) rely on iterative message passing between neighbouring nodes. While these models exhibit linear time complexity in the number of edges and are effective for small to moderately sized graphs, their scalability can be further enhanced. In particular, repeated aggregation steps across multiple layers may incur redundant computations and memory overhead when deployed on large-scale or densely connected graphs. Furthermore, their design typically emphasises fine-grained, local feature interactions at the node level, which often leads to over-fragmented representations that fail to capture long-range dependencies and global semantic structure, which are critical components in tasks requiring global graph understanding.

To address these limitations, we propose a new model called *Scalable Hierarchical Kirchhoff-Forest Graph Neural Network (SHAKE-GNN)*, a novel hierarchical graph neural architecture that constructs multi-resolution representations via a *Kirchhoff Forest-based* coarse-graining method. A spanning forest is a collection of spanning trees, each covering a connected component of a graph without forming cycles. Building on this, Kirchhoff Forests (KFs) are probabilistic ensembles of spanning forests, where the probability of each forest is determined by the graph Laplacian. This formulation provides a principled way to capture structural dependencies and connectivity patterns within graphs. Instead of relying on random or flat clustering schemes, we introduce a principled, layer-wise node merging strategy guided by Kirchhoff Forests (KFs), which generates a hierarchy of coarsened graphs. Each coarse node aggregates a substructure from the previous layer, using mean or sum pooling for node and edge features depending on the dataset.

Our main contributions are summarised as follows:

- We define a hierarchical graph decomposition pipeline based on KFs and integrate it with the model architecture SHAKE-GNN, which effectively leverages multi-resolution structural information for graph classification tasks. This framework is applicable across diverse graph domains and scales efficiently to large datasets.

- We introduce an improved strategy for selecting the resolution parameter $q$, balancing computational complexity and information loss, and provide a theoretical time complexity analysis that demonstrates the scalability advantages of our architecture over conventional GNN frameworks in multi-resolution settings.

- We empirically validate our approach through extensive experiments, showing that it achieves scalability compared to state-of-the-art GNN models.

## 2   RELATED WORK

**Graph Neural Networks (GNNs).** Graph Neural Networks (GNNs) have emerged as a powerful paradigm for representation learning on graph-structured data. GNNs function by recursively aggregating information from local neighbourhoods. The seminal Graph Convolutional Network (GCN) (Kipf & Welling, 2017) combines a node's features with those of its neighbours and calculates a weighted average over neighbourhoods. Subsequent extensions such as Graph Attention Networks (GAT) (Veličković et al., 2018) introduce adaptive weighting via an attention mechanism, while Graph Isomorphism Networks (GIN) (Xu et al., 2019) employ injective aggregation functions to enhance expressive capacity. These models have been successfully applied to a range of tasks. Typical examples include molecular property prediction (Hu et al., 2020), citation graph classification (Sen et al., 2008; Yang et al., 2016), and social network modeling (Hamilton et al., 2017; Li et al., 2020).

Despite their empirical success, conventional GNNs encounter significant limitations when scaled to large or densely connected graphs. Their reliance on multi-hop message passing increases computational overhead, particularly in deep architectures (Dwivedi et al., 2022). Moreover, their inherently local aggregation mechanisms often struggle to capture long-range dependencies and global semantics, resulting in representations that are focused on neighborhood-level patterns.

**Graph Pooling and Coarsening.** To mitigate the locality and scalability issues inherent in GNNs, various graph pooling and coarsening techniques have been developed to enable hierarchical representation learning. Differentiable Pooling (DiffPool) (Ying et al., 2018) introduces differentiable assignment matrices to softly cluster nodes into coarse representations, facilitating end-to-end hierarchical learning. Graph clustering using weighted cuts (Graclus) (Dhillon et al., 2007) adopts a greedy coarsening strategy based on graph cuts, while minimum-cut-based pooling (minCUT pooling) (Bianchi et al., 2020) imposes spectral regularisation to enforce partition quality.

Traditional graph partitioning algorithms, such as METIS (Karypis & Kumar, 1998), employ multilevel coarsening and refinement heuristics to minimize edge cuts while maintaining balanced partition sizes. This has enabled GNNs to scale to graphs with billions of nodes. A recent extension, LPMetis (Zeng et al., 2023), integrates label propagation into the coarsening phase to improve scalability and partition quality, particularly for distributed GNN training. Nevertheless, METIS and its variants are inherently non-stochastic.

**Personalised PageRank for Node-Level Scalability.** To mitigate the computational inefficiencies associated with recursive message passing in Graph Neural Networks (GNNs), recent research has proposed decoupling feature propagation from transformation via *Personalised PageRank* (PPR) scores. An illustrative instance of this approach is PPRGO (Bojchevski et al., 2020), which utilises a precomputed and sparsified approximation to facilitate efficient feature aggregation.

Despite their efficiency, PPR-based models such as PPRGO are node-centric in design. Each node's prediction is computed conditioned solely on its personalised neighbourhood. This localised perspective, while computationally advantageous, limits their applicability to graph-level tasks, such as molecular property prediction or program classification, where coarse-grained, hierarchical, or long-range interactions are significant.

## 3 PRELIMINARY

In this section, we introduce the fundamental concepts required for our framework. We begin by recalling classical notions of spanning trees and spanning forests, which serve as the basis of Kirchhoff Forests.

**Definition 3.1** (Spanning Tree). A *spanning tree* of $\mathcal{G}$ is a subgraph $\mathcal{T} = (\mathcal{V}_\mathcal{T}, \mathcal{E}_\mathcal{T})$ such that:

1. $\mathcal{T}$ includes all vertices of $\mathcal{G}$, i.e., $\mathcal{V}_\mathcal{T} = \mathcal{V}$;

2. $\mathcal{T}$ is acyclic, i.e., it contains no cycles;

3. $\mathcal{T}$ is connected, i.e., there exists a path between any two vertices in $\mathcal{T}$.

**Definition 3.2** (Spanning Forest). A *spanning forest* is a subgraph $\mathcal{F} = (\mathcal{V}, \mathcal{E}_\mathcal{F})$ such that:

1. Each connected component of $\mathcal{F}$ is a spanning tree of a connected component of $\mathcal{G}$;

2. $\mathcal{F}$ is acyclic and covers all vertices $\mathcal{V}$.

**Kirchhoff Forest-Based Graph Decomposition. (Bressan & Vigna, 2023; Barthelmé et al., 2025)** Kirchhoff Forests (KFs) offer a principled and probabilistically grounded framework for hierarchical graph decomposition. Rooted in spectral graph theory, KFs generate structured, multiscale partitions of nodes by sampling random spanning forests from a distribution derived from the graph Laplacian (Bressan & Vigna, 2023). The resolution of the decomposition is modulated by a temperature-like parameter $q > 0$, which controls the probabilistic distribution over forest structures and emulates fine-to-coarse abstraction.

Formally, let $\mathcal{G} = (\mathcal{V}, \mathcal{E})$ be an undirected graph. Kirchhoff Forests define a distribution over rooted spanning forests $\mathcal{F} \subseteq \mathcal{E}$ with root set $\mathcal{R} \subseteq \mathcal{V}$, governed by the parameter $q$. The probability of sampling a particular forest $\mathcal{F}$ is given by,

$$\mathbb{P}_q(\mathcal{F}) \propto \prod_{v \in \mathcal{V}} \left( \frac{q}{q + d_v} \right)^{\delta_v(\mathcal{F})} \prod_{(i,j) \in \mathcal{F}} \frac{1}{q + d_i},$$

where $d_v$ denotes the degree of node $v$, and $\delta_v(\mathcal{F})$ is an indicator function equal to 1 if $v$ is a root in $\mathcal{F}$, and 0 otherwise. This formulation induces a trade-off between the number and size of trees in the forest, allowing the parameter $q$ to control the expected number of connected components and the granularity of the resulting partition.

Sampling from this distribution is typically performed using Wilson's algorithm (Wilson, 1996), adapted with priority-based root selection to incorporate the influence of $q$. Priority-based root selection refers to a biasing mechanism whereby nodes are assigned selection priorities proportional to their restart probabilities. At each step in Wilson's algorithm, a node initiates a loop-erased random walk with stochastic restarts: it either terminates at a new root with probability $\frac{q}{q + d_v}$, or continues to a randomly chosen neighbour with probability $\frac{d_v}{q + d_v}$. The resulting collection of walks yields a spanning forest rooted at a dynamically constructed set of nodes.

To construct a hierarchical, multi-resolution decomposition, a strictly decreasing sequence of resolution parameters $q_1 > q_2 > \cdots > q_{N_q}$ is applied recursively. Each level produces increasingly fine-grained structural abstractions. To maintain consistency and reduce redundant computation across levels, the *Reboot* (Wilson, 1996) algorithm incrementally adjusts previously sampled forests by locally reassigning root nodes in response to a smaller resolution parameter $q' < q$. This approach preserves the probabilistic semantics of the distribution while avoiding full re-sampling at each stage of the hierarchy.

**Graph Hierarchy Construction.** We input a graph. Each node $v \in \mathcal{V}$ is associated with a feature vector in the node feature matrix $\mathbf{X}_v \in \mathbb{R}^{n \times f_v}$, where $f_v$ is the dimensionality of node attributes and $n$ is the number of nodes. Similarly, each edge $e \in \mathcal{E}$ is associated with a feature vector in the edge feature matrix $\mathbf{X}_e \in \mathbb{R}^{m \times f_e}$, where $f_e$ denotes the dimensionality of edge attributes and $m$ is the number of edges. The graph connectivity is represented by the adjacency matrix $\mathbf{A} \in \mathbb{R}^{n \times n}$, and the corresponding graph-level target label is denoted as $y \in \mathcal{Y}$.

To construct hierarchical representations suitable for multi-resolution graph classification, we pre-process each input graph using a coupled KF decomposition pipeline. Given a graph $\mathcal{G}$, we begin by extracting its structural and feature information, including node features, edge features, and connectivity. We then apply the Wilson algorithm to generate a collection of randomised spanning forests, each governed by a resolution parameter $q$ drawn from a strictly decreasing sequence $q_1, q_2, \ldots, q_{N_q}$. Each value of $q$ controls the distribution over forest structures, where larger values yield coarser partitions with fewer components, and smaller values induce finer-grained decompositions.

At each level of this hierarchy, nodes in the original graph are assigned to disjoint subsets, referred to as forest components, which form the basis for coarse nodes in the coarsened graph. Importantly, the features of each coarse node are obtained by directly averaging or summing the feature vectors of its constituent nodes from the original input graph, rather than from features from previously coarsened levels. Similarly, the edge attributes between coarse nodes are computed by averaging the features of all edges in the original graph that connect nodes across the corresponding components. By recursively applying this decomposition procedure across the sequence of $q$-values, we obtain a hierarchy of coarsened graphs, each capturing progressively higher-order structural abstractions.

For each pair of consecutive levels, we define a partition matrix $\mathbf{P}^{(\ell_{d-1}, \ell_d)}$ to specify and realize the coarse-graining operation between level $\ell_{d-1}$ and level $\ell_d$, where $\ell_d \in \{1, \ldots, N_q\}$. We extract the following components:

- The coarsened graph structure constructed by grouping nodes within each forest component into a supernode;

- Node features for each super-node computed by averaging or summing the feature vectors of constituent nodes in the original input graph;

- Edge features derived by averaging or summing the attributes of all original edges that connect nodes across corresponding forest components;

- Partition matrices $\mathbf{P}^{(\ell_{d-1}, \ell_d)} \in \mathbb{R}^{n_{\ell_{d-1}} \times n_{\ell_d}}$, where $n_{\ell_{d-1}}$ and $n_{\ell_d}$ denote the number of nodes at level $\ell_{d-1}$ and $\ell_d$.

It is important to note that the partition matrices $\mathbf{P}^{(\ell_{d-1}, \ell_d)}$ are not computed with respect to the original graph, but are instead derived recursively based on the coarsened structure from the preceding level. For example, if $\mathbf{P}^{(0,i)}$ and $\mathbf{P}^{(0,j)}$ denote mappings from the original graph to levels $i$ and $j$ respectively, then their product satisfies $\mathbf{P}^{(i,j)} = \mathbf{P}^{(0,j)} \big( \mathbf{P}^{(0,i)} \big)_R^{-1}$, illustrating the compositional consistency of the hierarchy. A detailed proof is presented in Appendix A. This recursive construction ensures consistent hierarchical alignment while accommodating the stochastic deviations introduced during random forest generation. Since nodes from different components in the finer level may be grouped into the same supernode at the coarser level, the resulting partition matrix naturally encodes soft associations as continuous values in $[0, 1]$, representing the proportion of contribution from each fine-level node.

This preprocessing yields a hierarchy of graphs $\{\mathcal{G}^{(0)}, \ldots, \mathcal{G}^{(N_q)}\}$ per graph, where $\mathcal{G}^{(0)} = \mathcal{G}$ denotes the original graph. The resulting hierarchical dataset is used to train our SHAKE-GNN model.

## 4 METHODOLOGY

In industry, graph classification in domains such as molecular property prediction and social network analysis requires models to be scalable. Conventional message-passing GNNs are limited by their local aggregation schemes and insufficient scalability, while pooling-based methods often rely on rigid heuristics. To address these issues, we propose *SHAKE-GNN*, a hierarchical architecture that employs *Kirchhoff Forests* for probabilistically grounded multi-resolution decomposition.

### 4.1 DATA-DRIVEN $q$-CHOICE VIA INFORMATION LOSS–COMPLEXITY TRADE-OFF

A central component of SHAKE-GNN is the selection of the smoothing parameter $q$, which governs the resolution of the Kirchhoff Forest decomposition. To determine an appropriate value of $q$, we adopt a data-driven strategy that balances information preservation against model complexity.

Building on the work of Tremblay et al. (2023), we introduce an improved framework. Specifically, we define Tikhonov smoothing operators on both nodes and edges, yielding smoothed features $\widehat{\mathbf{X}}_v = \mathbf{K}(q)\mathbf{X}_v$ and $\widehat{\mathbf{X}}_e = \mathbf{K}_e(q)\mathbf{X}_e$, where $\mathbf{K}(q) = q(\mathbf{L} + q\mathbf{I})^{-1}$ and $\mathbf{K}_e(q) = q(\mathbf{L}_e + q\mathbf{I})^{-1}$ are defined on the unnormalised graph Laplacian $\mathbf{L}$ and line-graph Laplacian $\mathbf{L}_e$, respectively. We measure information loss as the average of feature reconstruction error

$$\mathcal{L}_{\mathrm{rec}}(q) = \frac{\|\mathbf{X}_v - \widehat{\mathbf{X}}_v\|_F^2}{\lim_{q \to 0^+} \|\mathbf{X}_v - \widehat{\mathbf{X}}_v\|_F^2},$$

and Dirichlet energy loss

$$\mathcal{L}_{\mathrm{dir}}(q) = \frac{\mathrm{tr}((\mathbf{X}_v - \widehat{\mathbf{X}}_v)^\top \mathbf{L}(\mathbf{X}_v - \widehat{\mathbf{X}}_v))}{\mathrm{tr}(\mathbf{X}_v^\top \mathbf{L}\mathbf{X}_v)}.$$

This optimisation problem admits a more efficient spectral formulation. Instead of repeatedly computing matrix inverses for different values of $q$, one can perform a single eigendecomposition of $\mathbf{L}$ and $\mathbf{L}_e$, after which the quantities for all $q$ are obtained by simple per-eigenvalue evaluations. A detailed proof is presented in Appendix B.

Let $\mathbf{L} = \mathbf{U}\mathbf{\Lambda}\mathbf{U}^\top$ be the eigendecomposition of the Laplacian, with eigenvalues $\{\mu_i\}$ and eigenvectors $\{u_i\}$. In this basis, the Tikhonov gain is $h_i(q) = \frac{q}{\mu_i + q}$, and all quantities decompose into per-eigenmode contributions.

The node-side reconstruction error reduces to

$$\|\mathbf{X}_v - \widehat{\mathbf{X}}_v\|_F^2 = \sum_{i=1}^n (1 - h_i(q))^2 \|\mathbf{U}_i^\top \mathbf{X}_v\|_2^2,$$

while the Dirichlet energy loss becomes

$$\mathrm{tr}\big((\mathbf{X}_v - \widehat{\mathbf{X}}_v)^\top \mathbf{L}(\mathbf{X}_v - \widehat{\mathbf{X}}_v)\big) = \sum_{i=1}^n \mu_i (1 - h_i(q))^2 \|\mathbf{U}_i^\top \mathbf{X}_v\|_2^2.$$

Analogously, on the edge side we have

$$\|\mathbf{X}_e - \widehat{\mathbf{X}}_e\|_F^2 = \sum_{i=1}^m (1 - h_i^e(q))^2 \|\mathbf{U}_i^{e\top} \mathbf{X}_e\|_2^2,$$

and

$$\mathrm{tr}\big((\mathbf{X}_e - \widehat{\mathbf{X}}_e)^\top \mathbf{L}_e(\mathbf{X}_e - \widehat{\mathbf{X}}_e)\big) = \sum_{i=1}^m \mu_i^e (1 - h_i^e(q))^2 \|\mathbf{U}_i^{e\top} \mathbf{X}_e\|_2^2,$$

where $\mathbf{L}_e = \mathbf{U}_e \mathbf{\Lambda}_e \mathbf{U}_e^\top$ is the eigendecomposition of the line-graph Laplacian, and $h_i^e(q) = \frac{q}{\mu_i^e + q}$.

To quantify model complexity, we compute the effective degrees of freedom on both the node and edge sides. For the graph Laplacian $\mathbf{L}$ with eigenvalues $\{\mu_i\}_{i=1}^n$ and the line-graph Laplacian $\mathbf{L}_e$ with eigenvalues $\{\mu_i^e\}_{i=1}^m$, we define

$$\mathrm{df}_{\mathrm{node}}(q) = \frac{1}{n} \sum_{\mu_i > 0} \frac{q}{\mu_i + q} = \frac{1}{n} \sum_{\mu_i > 0} h_i(q),$$

$$\mathrm{df}_{\mathrm{edge}}(q) = \frac{1}{m} \sum_{\mu_i^e > 0} \frac{q}{\mu_i^e + q} = \frac{1}{m} \sum_{\mu_i^e > 0} h_i^e(q),$$

where $h_i(q) = \frac{q}{\mu_i + q}$ with eigenvalues $\{\mu_i\}$ of $\mathbf{L}$, and $h_i^e(q) = \frac{q}{\mu_i^e + q}$ with eigenvalues $\{\mu_i^e\}$ of the line-graph Laplacian $\mathbf{L}_e$. These spectral formulas enable efficient evaluation of $\mathcal{J}(q)$ and highlight its interpretation as a frequency-selective trade-off, where $q$ acts as a spectral filter modulating the contribution of each eigenmode.

The final objective combines node and edge information losses with a complexity penalty,

$$\mathcal{J}(q) = \mathcal{L}_{\mathrm{info,node}}(q) + \mathcal{L}_{\mathrm{info,edge}}(q) + \phi\big(\mathrm{df}_{\mathrm{node}}(q) + \mathrm{df}_{\mathrm{edge}}(q)\big),$$

where $\phi > 0$ controls the trade-off. The optimal resolution parameter is then selected as

$$q^\star = \arg\min_{q > 0} \ \mathcal{J}(q). \tag{1}$$

This formulation yields an interpretable criterion: small $q$ values preserve fine-grained information at the expense of complexity, while large $q$ reduce complexity but risk excessive information loss. The proposed trade-off identifies the resolution that best balances these effects for a given dataset.

## 4.2 SEQUENTIAL SHAKE-GNN ARCHITECTURE

The *Sequential SHAKE-GNN* architecture is designed to exploit the hierarchical inductive bias of multi-resolution graph decompositions by processing coarse-to-fine structural abstractions in a level-wise sequential fashion. Given an input graph $\mathcal{G}$, where each node and edge is endowed with a feature vector, the model propagates information through a progressive refinement pipeline that traverses a hierarchy of coarsened graphs.

Let $\mathbf{X}_v \in \mathbb{R}^{n \times f_v}$ and $\mathbf{X}_e \in \mathbb{R}^{m \times f_e}$ denote the initial node and edge feature matrices, respectively. These features are first embedded into a shared latent space $\mathbb{R}^o$ via dedicated encoders,

$$\mathbf{H}_v^{(0)} = \text{NodeEncoder}(\mathbf{X}_v), \quad \mathbf{H}_e^{(0)} = \text{EdgeEncoder}(\mathbf{X}_e).$$

Depending on the dataset, the encoders are instantiated either as embedding layers for categorical attributes or as multilayer perceptrons (MLPs) for continuous feature vectors.

The model iteratively processes the coarsened graphs corresponding to the resolution levels $q_1, \ldots, q_{N_q}$. At each resolution level $q_i$, a stack of $L_i$ message passing layers is applied, with updates at each layer $\ell \in \{1, \ldots, L_i\}$ defined as,

$$\mathbf{h}_i^{(\ell)} = \phi_{\text{msg}} \left( \mathbf{h}_i^{(\ell-1)}, \mathbf{h}_j^{(\ell-1)}, \mathbf{e}_{ij}^{(\ell-1)} \right),$$

where $\phi_{\text{msg}}$ denotes a message passing function involving trainable parameters, $\mathbf{h}_i^{(\ell)} \in \mathbb{R}^o$ denotes the hidden representation of node $i$ from the neural layer $\ell$, $\mathbf{e}_{ij}^{(\ell)} \in \mathbb{R}^o$ represents the edge feature associated with edge $(i,j)$, $j$ indexes the neighbors of node $i$ in the coarsened graph.

Upon completing the processing of message passing at level $q_i$, the resulting node embeddings $\mathbf{H}_v^{(L_i)}[q_i] \in \mathbb{R}^o$ are propagated to the next finer level $q_{i+1}$. Specifically, a stochastic alignment matrix $\mathbf{P}^{(i-1,i)} \in \mathbb{R}^{n_i \times n_{i-1}}$ encodes the distribution correspondence between coarse and fine level nodes,

$$\mathbf{H}_v^{(0)}[q_{i+1}] = \mathbf{P}^{(i-1,i)} \mathbf{H}_v^{(L_i)}[q_i].$$

After the GNN layers, a global mean pooling operator aggregates the node-level embeddings into a graph-level representation,

$$\mathbf{H}_{\text{comp}} = \text{GlobalMeanPool}(\mathbf{H}_v^{(L_{N_q})}[q_{N_q}]).$$

To further refine the representation, an optional read-out MLP composed of $L_{\text{MLP}}$ fully connected layers is applied,

$$\widehat{y} = \text{MLP}(\mathbf{H}_{\text{comp}}) \in \mathbb{R}^{o_y},$$

where $o_y$ denotes the output dimensionality corresponding to the number of classes.

## 4.3 TIME COMPLEXITY ANALYSIS

In this section, we provide a theoretical analysis of the computational complexity of SHAKE-GNN. We first estimate the expected size of the coarsened graphs produced by the Kirchhoff Forest decomposition, and then derive the overall time complexity of the model as a function of the resolution parameter $q$ and the number of resolution levels $N_q$.

### 4.3.1 EXPECTED COARSE GRAPH SIZE

To assess the computational efficiency of our model, it is essential to characterise how the graph size evolves under coarsening. Given an input graph $\mathcal{G}$, we estimate the expected size of the coarsened graph generated by applying the Kirchhoff Forest (KF) decomposition with resolution parameter $q$.

**Expected Number of Coarse Nodes.** In the Wilson-based forest sampling procedure, each node $v \in \mathcal{V}$ independently becomes the root of a tree with probability

$$p_v = \frac{q}{q + d_v},$$

where $d_v$ is the degree of node $v$. The expected number of coarse nodes (roots) is thus

$$\mathbb{E}[|\mathcal{V}_q|] = \sum_{v \in \mathcal{V}} \frac{q}{q + d_v}.$$

In our complexity calculation, assuming approximate uniform degree $d_v \approx \bar{d} = \frac{2m}{n}$, we obtain

$$\mathbb{E}[|\mathcal{V}_q|] \approx n \frac{q}{q + \bar{d}}.$$

**Expected Number of Coarse Edges.** An edge $(i, j) \in \mathcal{E}$ contributes to a coarse edge if it connects nodes belonging to different forest components. The probability of being cut can be calculated by (Bressan & Vigna, 2023; Barthelmé et al., 2025)

$$\mathbb{P}_{\mathrm{cut}}(i, j) = \frac{2q}{2q + d_i + d_j}.$$

Hence,

$$\mathbb{E}[|\mathcal{E}_q|] = \sum_{(i,j) \in \mathcal{E}} \frac{2q}{2q + d_i + d_j}.$$

With $d_i \approx d_j \approx \bar{d}$, this simplifies to (Bressan & Vigna, 2023)

$$\mathbb{E}[|\mathcal{E}_q|] \approx m \frac{q}{q + \bar{d}}.$$

The estimates of nodes and edges numbers show that both shrink by a resolution-dependent factor

$$r(q) = \frac{q}{q + \bar{d}},$$

so that

$$|\mathcal{V}_q| \approx r(q)\, n, \qquad |\mathcal{E}_q| \approx r(q)\, m.$$

### 4.3.2 TIME COMPLEXITY

For one resolution level $q$, the computational cost consists of:

- **Input Embedding.** $\mathcal{O}\big(r(q)\, n f_v o + r(q)\, m f_e o\big)$;

- **Message Passing and MLPs.** $\mathcal{O}\big(T\, r(q)(\, mo + nMo^2)\big)$;

- **Graph-Level Pooling.** $\mathcal{O}(r(q)\, no)$;

where $M$ denotes the depth of the internal multilayer perceptron (MLP), i.e., the number of fully connected layers within each message passing layer. Among these three components, the overall complexity is dominated by the term **Message Passing and MLPs**.

If the model employs $N_q$ distinct resolution levels, the total complexity is

$$\mathcal{O}\big(N_q\, r(q)\, T\big(mo + nMo^2\big)\big).$$

This formulation shows explicitly how the reduction ratio $r(q)$ controls the trade-off between efficiency and representation capacity across multiple resolutions. Since $r(q) < 1$, the total complexity of SHAKE-GNN is strictly lower than that of a standard GNN.

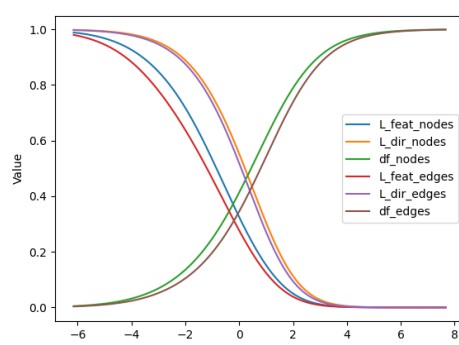 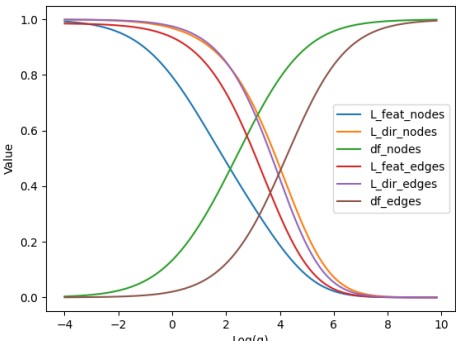

Figure 1: The $q$-choice figure of MOLHIV and PPA dataset, respectively. The six curves correspond to the node feature reconstruction loss, node structural loss, edge feature reconstruction loss, edge structural loss, as well as the node and edge complexity terms. Other figures in Appendix C.

## 5 RESULT

To validate the proposed SHAKE-GNN framework, we conduct comprehensive experiments designed to assess both predictive performance and computational efficiency. Our evaluation focuses on whether the incorporation of multi-resolution structures and Kirchhoff Forest-based coarsening can achieve competitive accuracy while significantly reducing training cost. In what follows, we first detail the experimental setup, then analyse the effect of resolution parameter selection, and finally present dataset-specific results with comparisons to standard baselines.

### 5.1 EXPERIMENTAL SETUP

To rigorously assess the effectiveness and generalizability of the proposed architecture, we conduct a series of graph classification experiments across multiple benchmark datasets. These datasets are drawn from diverse domains, including molecular chemistry and social network analysis. A description of the datasets is presented in Appendix D.

All experiments are conducted on a single NVIDIA RTX A6000 GPU equipped with 48GB of memory. The computational environment also includes an Intel Xeon W5-3425 CPU and 256GB of system RAM, providing sufficient resources for training efficient large-scale graph neural networks. The operating system is Ubuntu 24.04. The software stack comprises Python 3.11 and PyTorch 2.4.0, with CUDA version 12.4 for GPU acceleration.

### 5.2 RESOLUTION PARAMETER SELECTION

All experiments share a common set of training hyperparameters: batch size of 256, learning rate fixed at 0.005, weight decay set to $1.0 \times 10^{-5}$, random seed 42, and a maximum of 100 training epochs. A constant learning rate scheduler is used throughout. To prevent overfitting, we apply early stopping with patience of 10 epochs and a minimum improvement threshold of 0.001.

For each dataset, we perform a spectral evaluation of $\mathcal{J}(q)$ on the training split and compute the minimiser $q^\star$ defined in Equation 1. The resulting values are then applied throughout the training and evaluation phases. The result is shown in Figure 1.

By traversing a range of $q$ values, we obtain the combined objective $\mathcal{J}(q)$, whose minimum directly identifies the optimal resolution parameter $q^\star$ defined in Equation 1. This procedure systematically balances information preservation with computational efficiency, and similar evaluations are performed for all datasets.

In addition, each dataset uses specific architectural and optimization settings. These include the number of GNN layers, the number of linear layers per GNN block, the number of MLP layers in the

post-processing module, the hidden dimension size, and the optimiser. The complete configuration is summarised in Table 1. The grid search domain is presented in Appendix E.

Table 1: Dataset-specific hyperparameters used in experiments. INF indicates the original graph.

| Dataset | q Values | Layers per q Value | MLP Layers | Linear per GNN Layer | Hidden Dim | Optimizer |
|---|---|---|---|---|---|---|
| MolHIV | [INF, 1.9] | [6, 4] | 6 | 1 | 1024 | PESG |
| MolPPA | [INF, 207.0] | [4, 2] | 0 | 2 | 256 | AdamW |
| COLLAB | [INF, 17.8] | [4, 4] | 4 | 2 | 64 | AdamW |
| DD | [INF, 6.4] | [6, 4] | 4 | 1 | 64 | AdamW |
| REDDIT-MULTI-12K | [INF, 0.03] | [4, 4] | 0 | 2 | 128 | AdamW |

We compare the baseline GNN trained on the original graphs with our hierarchical variant using KF-based coarsening. For each dataset, we report both the training time (in minutes) and evaluation performance (ROC-AUC for molecular datasets, accuracy for social/protein datasets). Results are given for the original graphs and for the coarsened graphs, enabling direct comparison of efficiency and predictive power.

Table 2: Comparison of training time and evaluation performance between original and coarsened graphs.

| Dataset | Training Time (mins) | | Evaluation | |
|---|---|---|---|---|
| | Original | Coarsened | Original | Coarsened |
| MolHIV | 6.83 | 2.63 (38.53%) | 0.794 | 0.787 (99.11%) |
| MolPPA | 151.740 | 73.48 (48.42%) | 0.767 | 0.744 (97.00%) |
| COLLAB | 9.41 | 2.57 (27.30%) | 0.752 | 0.739 (98.27%) |
| DD | 0.24 | 0.09 (37.43%) | 0.749 | 0.740 (98.86%) |
| REDDIT-MULTI-12K | 5.45 | 1.99 (36.46%) | 0.492 | 0.481 (97.78%) |

Across multiple benchmark datasets, our experimental results consistently validate the effectiveness of incorporating multi-resolution structure and stochastic abstraction via the proposed Kirchhoff Forest-based graph coarsening framework. The SHAKE-GNN architecture achieves competitive or superior performance compared to standard GCN baselines, while at the same time significantly reducing training time in several configurations. Across all datasets, we achieved at least 97% of the baseline performance with the cost of at most 50% of the baseline.

These results further underscore the importance of architectural design choices. Allocating moderate depth to the coarse levels and optionally incorporating lightweight read-out MLPs helps to recover predictive capacity while preserving efficiency. In this way, SHAKE-GNN demonstrates that carefully tuned multi-resolution decomposition can simultaneously reduce computational burden in line with theoretical complexity estimates and maintain strong performance across diverse graph domains, thereby establishing itself as a principled and practical solution for scalable graph classification.

## 6 CONCLUSION

In this paper, we presented SHAKE-GNN, a scalable hierarchical graph neural network that couples message passing with a Kirchhoff-Forest-based multi-resolution decomposition. By explicitly modeling structure across resolutions and introducing a data-driven strategy for selecting the smoothing parameter $q$ via an information–complexity trade-off, SHAKE-GNN attains competitive accuracy while improving computational efficiency. Our analysis quantified how KF coarsening contracts graph size and, in turn, reduces the dominant computational terms, clarifying when multi-resolution processing is provably cheaper than a vanilla GNN. Empirically, across molecular and social benchmarks, SHAKE-GNN matched or surpassed strong baselines while yielding tangible training-time savings. We believe KF-guided hierarchical modelling provides a principled path toward scalable GNNs that preserve both local fidelity and global semantics.

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

## A   PROOF OF HIERARCHICAL PARTITION CONSISTENCY

**Proposition A.1.** *For any hierarchy levels $i < j$, the partition matrices satisfy*

$$\mathbf{P}^{(i,j)} = \mathbf{P}^{(0,j)}\big(\mathbf{P}^{(0,i)}\big)_R^{-1}.$$

*Proof.* By definition, the partition matrix $\mathbf{P}^{(0,i)}$ maps the original feature matrix $\mathbf{F}^{(0)}$ at level 0 to its coarsened representation $\mathbf{F}^{(i)}$ at level $i$, i.e.,

$$\mathbf{P}^{(0,i)}\mathbf{F}^{(0)} = \mathbf{F}^{(i)}, \qquad \mathbf{P}^{(0,j)}\mathbf{F}^{(0)} = \mathbf{F}^{(j)}.$$

Similarly, the partition $\mathbf{P}^{(i,j)}$ maps between consecutive coarse levels,

$$\mathbf{P}^{(i,j)}\mathbf{F}^{(i)} = \mathbf{F}^{(j)}.$$

Combining these relations, we obtain

$$\mathbf{P}^{(i,j)}\mathbf{P}^{(0,i)}\mathbf{F}^{(0)} = \mathbf{P}^{(0,j)}\mathbf{F}^{(0)}.$$

Since this holds for arbitrary $\mathbf{F}^{(0)}$, it follows that

$$\mathbf{P}^{(i,j)}\mathbf{P}^{(0,i)} = \mathbf{P}^{(0,j)}.$$

Since $\mathbf{P}^{(0,i)}$ is generally a non-square partition matrix, its transpose is not a true inverse. Instead, we consider its *right inverse*, defined as

$$\big(\mathbf{P}^{(0,i)}\big)_R^{-1} = \mathbf{P}^{(0,i)\top}\big(\mathbf{P}^{(0,i)}\mathbf{P}^{(0,i)\top}\big)^{-1},$$

which satisfies $\mathbf{P}^{(0,i)}\big(\mathbf{P}^{(0,i)}\big)_R^{-1} = \mathbf{I}$. Substituting this into the relation yields

$$\mathbf{P}^{(i,j)} = \mathbf{P}^{(0,j)}\big(\mathbf{P}^{(0,i)}\big)_R^{-1}.$$

$\square$

## B   FROM THE INVERSE FORMULATION TO THE SPECTRAL FORMULATION

### B.1   NOTATION AND ASSUMPTIONS

Let $\mathbf{L} \in \mathbb{R}^{n \times n}$ be an unnormalised graph Laplacian, which is symmetric positive semidefinite. Let $\mathbf{L} = \mathbf{U}\mathbf{\Lambda}\mathbf{U}^\top$ be an eigen-decomposition, where $\mathbf{U} \in \mathbb{R}^{n \times n}$ is orthogonal, and $\mathbf{\Lambda} = \mathrm{diag}(\mu_1, \ldots, \mu_n)$ with $0 = \mu_1 \leq \cdots \leq \mu_n$. Given node features $\mathbf{X}_v \in \mathbb{R}^{n \times f_v}$ and a regularisation parameter $q > 0$, define the Tikhonov smoothing operator

$$\mathbf{K}(q) = q(\mathbf{L} + q\mathbf{I})^{-1}.$$

We write the smoothed signal as $\widehat{\mathbf{X}}_v = \mathbf{K}(q)\mathbf{X}_v$, and the residual as $\mathbf{R} = \mathbf{X}_v - \widehat{\mathbf{X}}_v$. For any matrix $\mathbf{M}$, $\|\mathbf{M}\|_F$ denotes the Frobenius norm and $\mathrm{tr}(\mathbf{M})$ the trace.

### B.2   SMOOTHING OPERATOR IN THE SPECTRAL BASIS

**Lemma B.1** (Spectral form of the resolvent)**.** *For any $q > 0$,*

$$(\mathbf{L} + q\mathbf{I})^{-1} = \mathbf{U}(\mathbf{\Lambda} + q\mathbf{I})^{-1}\mathbf{U}^\top.$$

*Proof.* Since $\mathbf{L} = \mathbf{U}\mathbf{\Lambda}\mathbf{U}^\top$ with $\mathbf{U}$ orthogonal,

$$\mathbf{L} + q\mathbf{I} = \mathbf{U}(\mathbf{\Lambda} + q\mathbf{I})\mathbf{U}^\top.$$

Taking the inverse on both sides and using $\mathbf{U}^{-1} = \mathbf{U}^\top$ yields the claim. $\square$

**Lemma B.2** (Spectral form of the smoother)**.** *Let $h_i(q) := \frac{q}{\mu_i + q}$. Then*

$$\mathbf{K}(q) = \mathbf{U}\,\mathrm{diag}\big(h_i(q)\big)\,\mathbf{U}^\top.$$

*Proof.* By Lemma B.1,

$$\mathbf{K}(q) = q(\mathbf{L} + q\mathbf{I})^{-1} = q\,\mathbf{U}(\mathbf{\Lambda} + q\mathbf{I})^{-1}\mathbf{U}^\top = \mathbf{U}\operatorname{diag}\!\left(\tfrac{q}{\mu_i+q}\right)\mathbf{U}^\top.$$

$\square$

Let $\widetilde{\mathbf{X}}_v := \mathbf{U}^\top \mathbf{X}_v$ be the spectral coefficients and define per-mode energies for $i = 1, \dots, n$,

$$S_i := \|\widetilde{\mathbf{X}}_{v,i:}\|_2^2 = \sum_{j=1}^{f_v} \widetilde{\mathbf{X}}_{v,ij}^2.$$

By Lemma B.2,

$$\widehat{\mathbf{X}}_v = \mathbf{K}(q)\mathbf{X}_v = \mathbf{U}\operatorname{diag}(h_i(q))\,\mathbf{U}^\top\mathbf{X}_v \;\Rightarrow\; \widehat{\widetilde{\mathbf{X}}}_v = \mathbf{U}^\top\widehat{\mathbf{X}}_v = \operatorname{diag}(h_i(q))\,\widetilde{\mathbf{X}}_v.$$

Therefore, the spectral residual is

$$\widetilde{\mathbf{R}} = \widetilde{\mathbf{X}}_v - \widehat{\widetilde{\mathbf{X}}}_v = \operatorname{diag}(1 - h_i(q))\,\widetilde{\mathbf{X}}_v.$$

### B.3 FEATURE RECONSTRUCTION LOSS: EQUIVALENCE

**Proposition B.3.** *Let $\mathbf{L} \in \mathbb{R}^{n\times n}$ be the graph Laplacian with eigendecomposition $\mathbf{L} = \mathbf{U\Lambda U}^\top$, where $\mathbf{U}$ is orthogonal and $\mathbf{\Lambda} = \operatorname{diag}(\mu_1, \dots, \mu_n)$. For node features $\mathbf{X}_v \in \mathbb{R}^{n\times f_v}$ and smoothing parameter $q > 0$, define the Tikhonov smoothing operator*

$$\mathbf{K}(q) = q(\mathbf{L} + q\mathbf{I})^{-1}, \qquad \widehat{\mathbf{X}}_v = \mathbf{K}(q)\mathbf{X}_v.$$

*Let $\widetilde{\mathbf{X}}_v = \mathbf{U}^\top \mathbf{X}_v$ be the spectral coefficients and $S_i = \|\widetilde{\mathbf{X}}_{v,i:}\|_2^2 = \sum_{j=1}^{f_v} \widetilde{\mathbf{X}}_{v,ij}^2$ the per-mode energies. With the spectral gain $h_i(q) = \frac{q}{\mu_i+q}$, the feature reconstruction loss admits the exact spectral form*

$$\|\mathbf{X}_v - \widehat{\mathbf{X}}_v\|_F^2 = \sum_{i=1}^n (1 - h_i(q))^2\,S_i, \qquad \|\mathbf{X}_v\|_F^2 = \sum_{i=1}^n S_i. \tag{2}$$

### B.4 DIRICHLET LOSS: EQUIVALENCE

**Proposition B.4.** *Let $\mathbf{L} \in \mathbb{R}^{n\times n}$ be the graph Laplacian with eigendecomposition $\mathbf{L} = \mathbf{U\Lambda U}^\top$, where $\mathbf{U}$ is orthogonal and $\mathbf{\Lambda} = \operatorname{diag}(\mu_1, \dots, \mu_n)$. For node features $\mathbf{X}_v \in \mathbb{R}^{n\times f_v}$ and smoothing parameter $q > 0$, define the Tikhonov smoothing operator*

$$\mathbf{K}(q) = q(\mathbf{L} + q\mathbf{I})^{-1}, \qquad \widehat{\mathbf{X}}_v = \mathbf{K}(q)\mathbf{X}_v.$$

*Let $\widetilde{\mathbf{X}}_v = \mathbf{U}^\top \mathbf{X}_v$ be the spectral coefficients and $S_i = \|\widetilde{\mathbf{X}}_{v,i:}\|_2^2 = \sum_{j=1}^{f_v} \widetilde{\mathbf{X}}_{v,ij}^2$ the per-mode energies. With the spectral gain $h_i(q) = \frac{q}{\mu_i+q}$ and residual $\mathbf{R} = \mathbf{X}_v - \widehat{\mathbf{X}}_v$, the Dirichlet energy loss admits the exact spectral form*

$$\operatorname{tr}\!\big((\mathbf{X}_v - \widehat{\mathbf{X}}_v)^\top\mathbf{L}(\mathbf{X}_v - \widehat{\mathbf{X}}_v)\big) = \sum_{i=1}^n \mu_i(1 - h_i(q))^2\,S_i, \qquad \operatorname{tr}(\mathbf{X}_v^\top\mathbf{L}\mathbf{X}_v) = \sum_{i=1}^n \mu_i S_i. \tag{3}$$

### B.5 EDGE/LINE-GRAPH CASE

Let $\mathbf{L}_e = \mathbf{U}_e\mathbf{\Lambda}_e\mathbf{U}_e^\top$ be the line-graph Laplacian and $\mathbf{X}_e \in \mathbb{R}^{m\times f_e}$ the edge features (optionally aggregated to undirected edges). Define $h_i^e(q) = \frac{q}{\mu_i^e+q}$, $\widetilde{\mathbf{X}}_e = \mathbf{U}_e^\top \mathbf{X}_e$, and $S_i^e = \|\widetilde{\mathbf{X}}_{e,i:}\|_2^2$. Then, repeating the node-side derivations verbatim,

$$\|\mathbf{X}_e - \widehat{\mathbf{X}}_e\|_F^2 = \sum_{i=1}^m (1 - h_i^e(q))^2\,S_i^e, \qquad\qquad \|\mathbf{X}_e\|_F^2 = \sum_{i=1}^m S_i^e,$$

$$\operatorname{tr}\!\big((\mathbf{X}_e - \widehat{\mathbf{X}}_e)^\top\mathbf{L}_e(\mathbf{X}_e - \widehat{\mathbf{X}}_e)\big) = \sum_{i=1}^m \mu_i^e(1 - h_i^e(q))^2\,S_i^e, \qquad \operatorname{tr}(\mathbf{X}_e^\top\mathbf{L}_e\mathbf{X}_e) = \sum_{i=1}^m \mu_i^e S_i^e,$$

$$\operatorname{tr}\mathbf{K}_e(q) = \sum_{i=1}^m h_i^e(q).$$

## C    ADDITIONAL $q$-CHOICE FIGURES

In the main text (Figure 1), we reported the $q$-choice curves for the OGBG-MOLHIV and OGBG-PPA datasets. For completeness, we provide here the corresponding plots for the remaining datasets, that are COLLAB, DD, and REDDIT-MULTI-12K. For datasets without edge features, the corresponding edge-related quantities remain constant at zero.

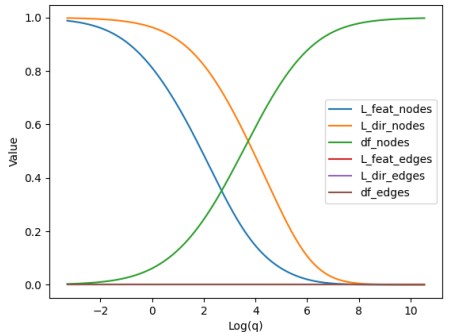

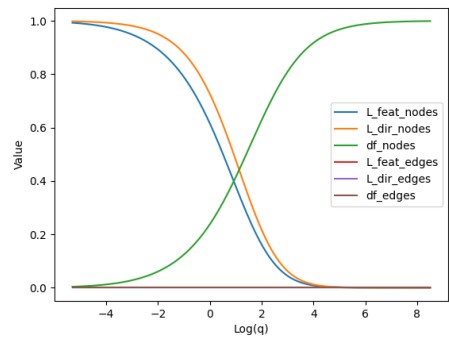

(a) The $q$-choice curves for the COLLAB dataset.

(b) The $q$-choice curves for the DD dataset.

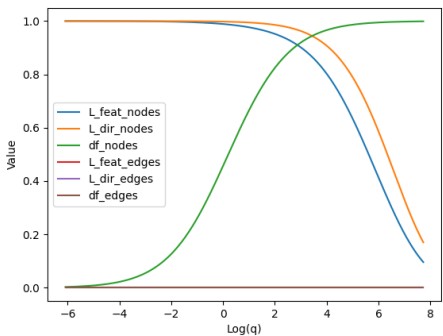

(c) The $q$-choice curves for the REDDIT-MULTI-12K dataset.

Figure 2: The $q$-choice curves for COLLAB, DD, and REDDIT-MULTI-12K datasets.

## D    EXPERIMENT DATASETS

**OGBG-MolHIV.**    The `ogbg-molhiv` dataset is a molecular graph classification benchmark originating from the MoleculeNet collection (Wu et al., 2018) and integrated into the Open Graph Benchmark (OGB) (Hu et al., 2020). Each sample is a molecule represented as a graph, with atoms as nodes and chemical bonds as edges. The prediction task is binary classification, determining whether a molecule exhibits HIV-inhibitory activity, evaluated using the ROC-AUC metric.

**OGBG-MolPPA.**    The `ogbg-ppa` dataset is a protein–protein association network from the OGB suite. Nodes correspond to proteins and edges indicate biological associations such as functional interactions. The task is graph classification, where the objective is to predict the biological class of a protein association graph, providing a challenging benchmark due to the large graph sizes and heterogeneity.

**COLLAB.**    The `COLLAB` dataset is a social network graph classification benchmark derived from scientific collaboration networks. Each graph corresponds to the ego-network of a researcher, where

nodes represent collaborators and edges represent co-authorship. The classification task is multi-class, aiming to predict the scientific field of the researcher based on the collaboration structure.

**DD.** The `DD` dataset (D&D) is a protein graph classification benchmark. Each graph corresponds to a protein, where nodes represent amino acids and edges are formed between pairs of amino acids that are spatially close in the 3D structure. The task is binary classification, predicting whether a protein is an enzyme or a non-enzyme.

**REDDIT-MULTI-12K.** The `REDDIT-MULTI-12K` dataset is a social network benchmark derived from Reddit discussion threads. Each graph represents a discussion thread, where nodes are users and edges indicate interactions (e.g., replies). The classification task is multi-class, predicting the category of the discussion thread (e.g., technology, politics, sports).

## E   HYPERPARAMETER GRID SEARCH DOMAIN

For reproducibility, we document the hyperparameter search domain used for tuning SHAKE-GNN and baseline models. The search was conducted via grid search over the following ranges:

- **Number of GNN layers per resolution level:** $\{2, 4, 6, 8, 10, 12\}$.
- **Read-out layers:** $\{0, 2, 4, 6, 8, 10, 12\}$.
- **Linear layers per GNN layer:** $\{1, 2, 3, 4\}$.
- **Hidden dimension:** $\{64, 128, 256, 512, 1024, 2048\}$.

