# OpenReview forum: "SHAKE-GNN: Scalable Hierarchical Kirchhoff-Forest Graph Neural Network"
_ICLR.cc/2026/Conference — Submitted to ICLR 2026_

### Official Review · Reviewer_TgSH · 2025-10-27

**Soundness:** 2
**Presentation:** 3
**Contribution:** 2
**Rating:** 4
**Confidence:** 4

**Summary:**

This paper studies large-scale graph-level tasks. SHAKE-GNN is proposed, which is a subgraph sampling method using Kirchoff Forests. A data-driven strategy is used for selecting the trade-off parameter.

**Strengths:**

S1: The writing is clear.

S2: The method is easy-to-follow.

S3: Hyperparameters are listed clearly.

**Weaknesses:**

W1: The motivation is not clear. Using subgraph sampling/coarsening/sparsification/condensation is common in scalable gnn, but the motivation of using Kirchhoff Forests for graph-level task is unclear.

W2: Some statements need formal proof. See questions.

W3: Experiments require considerable effort to improve.

**Questions:**

Q1: Why use Kirchhoff Forests for graph-level task? Is there any special reason that it will work better on graph-level tasks than standard node-classification?

Q2: Line 377, this statement is made: "Since $r(q)<1$, the total complexity of SHAKE-GNN is strictly lower than that of a standard GNN". Can you provide a formal proof on that? This does not seem trivial.

Q3: This paper studies large-scale graph-level tasks, but the datasets used in the experiments are small. The average number of node per graph is only 25.5 for the MolHIV datasets. Can you also provide statistics of each datasets as a table?

Q4: No proper baselines are selected. The proposed method is only compared with the vanilla GCN. Scalable graph learning methods should be added as baselines.

Q5: Is there setting difference between this paper and the official ogb leaderboard? It seems like lots of works achieve 80+ score in early 2021.

---

> ### Author Response · Authors · 2025-11-28
> **Reply**
>
> We sincerely thank the reviewer for their comments. Below we attempt to answer the posed questions in turn.
>
> 1. We thank the reviewer for this question. The rationale of using the Kirchhoff Forests construction specifically in graph-level instead of node-level tasks is that in the majority of the existing graph-level methodology, each node is accurately represented throughout much of the existing architectures at a significant computational expense. We see Kirchhoff Forests as a method that allows us to reduce the computational expense that is incurred in the representations of nodes, while keeping the performance cost manageable. We believe our SHAKE-GNN to offer an attractive, tunable tradeoff between scalability and accuracy. For node-level tasks, this rationale does not apply. We furthermore point out that for node-level tasks, there already exist high-performing, scalable GNN architectures, such as the PPRGo architecture. Contrariwise, for graph-level tasks, there currently exists a lack of scalable graph representation learning methodology. The contribution of our work is to fill this gap.
>
> 2. We thank the reviewer for pointing out that this statement requires a more careful justification. In the revised manuscript, we will provide a detailed derivation in the appendix. Specifically, we will first derive the computational complexity of a standard GNN layer in terms of the number of nodes, edges, hidden dimension, and number of layers. We will then derive the complexity of SHAKE-GNN under the same setting and show that the two expressions differ only by a multiplicative factor $r(q)$, which denotes the effective coarsening ratio induced by our scheme. By construction, $r(q)$ is strictly smaller than 1, which implies that the total complexity of SHAKE-GNN is strictly lower than that of the corresponding standard GNN with the same backbone architecture.
>
> 3. We thank the reviewer for this helpful suggestion. We agree that providing detailed dataset statistics will improve the clarity of the experimental section. In the revised manuscript, we will add a table in the appendix summarising key statistics for each dataset, including the number of graphs, average number of nodes, average number of edges, and task type. We will also include a brief comment explaining that, despite our focus on large-scale graph-level tasks, widely used benchmark datasets in this domain tend to have relatively small graph sizes. This contextual clarification will be added alongside the table in the appendix.
>
> 4. We agree that including comparisons with hierarchical GNNs, graph coarsening methods, and scalability-oriented approaches would significantly strengthen the empirical evaluation. In the revised version, we will add experiments with PPRGO, DiffPool, and METIS. These additional baselines will enable a more comprehensive and fair comparison across the relevant methodological categories.
>
> 5. We thank the reviewer for raising this point. We acknowledge that results reported on the official OGB leaderboard can vary depending on training settings, data augmentation strategies, and hyperparameter choices. To address this concern, we will conduct additional experiments using the official OGB settings to ensure full comparability with leaderboard results. The updated results and a clarification of any setting differences will be included in the revised manuscript.

---

### Official Review · Reviewer_PVGT · 2025-11-01

**Soundness:** 3
**Presentation:** 3
**Contribution:** 3
**Rating:** 6
**Confidence:** 2

**Summary:**

This paper studies the problem of scaling graph neural networks for large graph classification tasks and proposes a hierarchical framework based on Kirchhoff random forests. The method constructs multi-resolution graph representations by progressively coarsening the graph using probabilistically grounded Kirchhoff Forests, and performs message passing from coarse to fine levels. Experiments on multiple real-world datasets show that SHAKE-GNN significantly reduces training time while maintaining the performance of the original model.

**Strengths:**

1. The idea of Kirchhoff Forest provides a theoretically grounded way to generate multi-resolution graph structures, which can potentially inspire future studies.
2. The proposed strategy for choosing the parameter q is interesting and allows the framework adaptive to different datasets.

**Weaknesses:**

I don’t have too many questions regarding the methodology but have some concerns on the experiments:

1. All experiments are conducted using GCN backbones and it is unclear how this framework will perform across different GNN variants.
2. The experiments are primarily conducted on standard graph classification benchmarks; there is no evaluation on more challenging or large real-world industrial graphs, nor on other tasks such as graph regression or link prediction.

**Questions:**

See weaknesses.

---

> ### Author Response · Authors · 2025-11-28
>
> We very much thank you for your review. In the following, respond to the weaknesses you highlighted in turn.
>
> 1. We thank the reviewer for highlighting this point. We agree that evaluating our framework with different GNN backbones would provide a clearer picture of its generality. In the revised manuscript, we will include additional experiments using GAT in place of GCN to assess whether our method remains effective across different GNN architectures. These results will be added to the updated experimental section.
>
> 2. We thank the reviewer for this valuable comment. We agree that evaluating on more challenging real-world industrial graphs and on additional tasks such as graph regression or link prediction would further broaden the applicability of our study. However, most publicly available graph regression datasets consist of small molecular graphs, which fall outside the scope of our work, as our goal is to demonstrate scalability on large graph-level problems rather than on small-molecule settings. That said, we will include an additional graph regression experiment in the revised manuscript to provide broader task coverage. Moreover, we fully agree that access to large real-world industrial graphs would enable a much richer evaluation. We have therefore added this direction to the “Future Work” section as a meaningful extension of our current framework.

---

### Official Review · Reviewer_DUSA · 2025-11-01

**Soundness:** 2
**Presentation:** 2
**Contribution:** 2
**Rating:** 2
**Confidence:** 4

**Summary:**

This paper proposes SHAKE-GNN, a hierarchical graph neural network architecture that leverages Kirchhoff Forest (KF) decomposition for multi-resolution graph representation learning. The method constructs a hierarchy of coarsened graphs by sampling random spanning forests with varying resolution parameters q, then applies message passing at each level.

**Strengths:**

1. The paper correctly identifies the limitations of message-passing GNNs on large graphs and situates SHAKE-GNN within this context.

2. Employing KFs for hierarchical decomposition is an interesting idea grounded in spectral graph theory, providing a theoretically interpretable framework for stochastic multiresolution representation.

3. The analytical derivation of expected node/edge counts and asymptotic runtime are comprehensive.

**Weaknesses:**

1. The technical novelty is modest. The paper heavily builds on existing Kirchhoff-Forest decomposition methods: Bressan & Vigna (2023) and Barthelmé et al. (2025). Tikhonov smoothing for q-selection builds on Tremblay et al. (2023). The only distinct contribution is the data-driven q-selection, which, while useful, is an incremental refinement rather than a conceptual breakthrough.

2. I would suggest the authors clearly mention the motivation for the proposed design, even though the techniques are borrowed from other references. For example, why k(q) are defined like that on line 218? Is this from a Graph Signal Denoising problem?

3. Another concern is about the complexity. As mentioned, "one can perform a single eigen-decomposition of L and Le, after which the quantities for all q are obtained by simple per-eigenvalue evaluations." However, the Laplacian decomposition will have an approximate cubic complexity with respect to the number of nodes at worst, which is unrealistic to implement in practice. And this part is not mentioned in the complexity analysis section.

4. Following the previous point, empirical efficiency evaluation is necessary, especially on large-scale graphs.

5. The major problem is the evaluation of the paper. The authors compare only against a vanilla baseline GNN trained on original graphs. They completely fail to compare against:

(1) Other hierarchical GNN methods (DiffPool, minCUT pooling, SAGPool)

(2) Graph coarsening baselines (METIS, which they cite in related work)

(3) Other scalability-focused methods (PPRGO, GraphSAINT, ClusterGCN)

6. Since this coarsening is stochastic, the results in the table only have one single value, which is not promising, and the reproducibility is not clear. The mean and standard deviation are required.

7. Despite the "scalable GNN" claim, SHAKE-GNN is only evaluated on graph-classification benchmarks. There are no results on node classification, link prediction tasks, which are crucial to demonstrate scalability and generality.

8. The study also lacks analysis of how each component contributes to performance.

**Questions:**

Pease refer to the weakness

---

> ### Author Response · Authors · 2025-11-28
>
> We want to thank the reviewer very much for the time they dedicated to our work to produce their review. Below we provide our perspective on the raised weaknesses point-by-point.
>
> 1. Our primary contribution lies in the design of a specialised GNN architecture that leverages the Kirchhoff Forest construction to match the performance of baseline GNNs in a fraction of their computational time. This component is not merely an incremental addition: it achieves consistently strong empirical performance and demonstrates a clear advantage. We therefore believe the paper offers a meaningful step forward for the literature, complementing prior theoretical tools with a data-driven mechanism that yields state-of-the-art performance.
>
> 2. We agree that providing clearer motivation for the proposed design choices — particularly the definition of $K(q)$ and its connection to graph-signal denoising — will improve the clarity of the paper. Accordingly, we will add a concise explanation of $K(q)$ to the revised manuscript, including the rationale behind its formulation. We will also incorporate an intuitive description of the node-level quantity $\mathrm{df}_{node}$ to clarify its role in our method.
>
> 3. We agree that the need for a Laplacian eigendecomposition represents a genuine computational weakness of our current $q$-selection scheme. As noted, the worst-case complexity can indeed be cubic in the number of nodes. However, we emphasise that this step is performed only once as a preprocessing stage, and in practice, the sparsity of the Laplacian matrices allows significantly faster algorithms (as discussed in the Kirchhoff–Forest literature). Moreover, a variety of established techniques exist for further accelerating eigendecompositions on large sparse graphs. Since improving eigendecomposition speed is a broad and well-studied problem in its own right, we believe it falls outside the main scope of our work.
>
> 4. In selecting our benchmark datasets, we intentionally chose the largest commonly used graph-level datasets available in our literature. Unfortunately, we were unable to identify larger standard benchmarks. We would be very happy to extend our empirical evaluation to larger datasets to further demonstrate the efficiency of our method. If the reviewer is aware of such datasets, we would be grateful for suggestions.
>
> 5. We agree that including comparisons with hierarchical GNNs, graph coarsening methods, and scalability-oriented approaches would significantly strengthen the empirical evaluation. In the revised version, we will add experiments with PPRGO, DiffPool, and METIS. These additional baselines will enable a more comprehensive and fair comparison across the relevant methodological categories.
>
> 6. Since our coarsening procedure is stochastic, we agree that reporting only a single value is insufficient. In the revised manuscript, we will include mean and standard deviation over multiple runs to ensure reproducibility and provide a more reliable evaluation. These results will be added to the updated experimental section.
>
> 7. Our intention is not to propose a general-purpose scalable method for all graph-learning tasks. Rather, our focus is specifically on addressing the gap that exists in scalable GNNs for graph-level prediction. For node classification and link prediction, there already exist highly effective and scalable approaches such as PPRGo. In contrast, scalable solutions for graph-level tasks remain much more limited, and this is precisely the problem our SHAKE-GNN framework aims to address. We will clarify this scope and motivation more explicitly in the revised manuscript.
>
> 8. While we agree that ablation studies are standard in our community, our method is intentionally simple and built from a small number of well-motivated components. Because the approach does not combine multiple loosely connected modules, we do not believe that removing individual elements would yield ablations that are meaningful or informative, and we prefer to avoid adding experiments that would risk producing only trivial conclusions. That said, we would be grateful if the reviewer could point to any specific component for which they believe an ablation would provide substantive insight. If such a component is identified, we would be happy to incorporate the corresponding analysis in the revised manuscript.

---

### Official Review · Reviewer_Kxh4 · 2025-11-06

[review text omitted: it was posted to a different submission]

---

> ### Author Response · Authors · 2025-11-12
>
> Dear Reviewer Kxh4,
>
> The submitted review regrettably discusses a different paper than the manuscript submitted by us. In case our manuscript was reviewed by you, we would be very happy to have your comments on our work. Would it be possible for you to edit your review to include your comments as soon as possible, so we can formulate our answers in a timely manner?
>
> Thank you very much for the time you devote to our work!
>
> Kind regards,
>
> The Authors

---

### Meta-Review · Area_Chair_WUCY · 2025-12-30

**Summary:**

In this paper, the authors propose SHAKE-GNN, a scalable graph-level GNN based on a hierarchy of Kirchhoff Forests, which produces multi-scale representations and enables flexible trade-offs between efficiency and performance. Experimental results show the effectiveness of the proposed method while maintaining scalability.

The reviewers generally acknowledge that this paper provides an interesting perspective for large-scale GNNs. However, they raised several valid concerns, particularly with insufficient experiments (lacking large-scale benchmarks and numerous baselines), the rationale for the model design, and complexity concerns, among others. The authors acknowledge many of these points and agree to extend the experiments, but do not provide updated results. Overall, I think the paper provides a potentially promising angle for scalable GNNs, but many details, particularly experimental verifications, need more effort. In its current shape, the paper is clearly below the acceptance bar, and thus I recommend rejection.

Note: the comments of Reviewer Kxh4 are likely for another paper, and the score is omitted.

**Reviewer Concerns:**

For Reviewer Kxh4: the comments are likely for another paper, and the score is omitted.

For Reviewer DUSA:
W1/W2: Technical novelty/motivation.
The authors provided a response, but I doubt it can convince the reviewer.

W3: Another concern is about the complexity.
The authors provided a response, but I agree with the reviewer, and even one-time preprocessing eigen-decomposition is likely a bottleneck for processing large graphs.

W4: Empirical efficiency evaluation is necessary, especially on large-scale graphs.
The concern remains valid.

W5: The major problem is the evaluation of the paper.
The concern remains valid.

W6: Reproducibility is not clear.
The concern remains valid.

W7: There are no results on node classification, link prediction tasks.
The authors provided a response, but I doubt it can convince the reviewer.

W8: The study also lacks analysis of how each component contributes to performance.
The authors provided a response, but I doubt it can convince the reviewer.


For Reviewer PVGT:
W1: It is unclear how this framework will perform across different GNN variants.
The concern remains valid.

W2: There is no evaluation on more challenging or large real-world industrial graphs...
The concern remains valid.

For Reviewer TgSH:
W1: The motivation is not clear.
The authors provided a response, but I doubt it can convince the reviewer.

W2: Some statements need formal proof.
The rebuttal should address some of the concerns.

W3: Experiments require considerable effort to improve.
The concern remains valid.

**Reviewer Scores:**

For Reviewer Kxh4, the comments are likely for another paper and the score is omitted.

For Reviewer DUSA, the initial rating is 2, and it is likely to stay at 2.

For Reviewer PVGT, the initial rating is 6, and it is likely to stay at 6.

For Reviewer TgSH, the initial rating is 4, and it is likely to stay at 4.

---

### Decision · Program_Chairs · 2026-01-26

Reject